# Proinsulin C-Peptide Enhances Cell Survival and Protects against Simvastatin-Induced Myotoxicity in L6 Rat Myoblasts

**DOI:** 10.3390/ijms20071654

**Published:** 2019-04-03

**Authors:** Sumia Mohamed Essid, Alan Bevington, Nigel J. Brunskill

**Affiliations:** Department of Infection, Immunity and Inflammation, University of Leicester, Leicester LE1 7 RH, UK; ab74@le.ac.uk (A.B.); njb18@leicester.ac.uk (N.J.B.)

**Keywords:** proinsulin C-peptide, Type 1 diabetes, simvastatin, myotoxicity

## Abstract

The repair capacity of progenitor skeletal muscle satellite cells (SC) in Type 1 diabetes mellitus (T1DM) is decreased. This is associated with the loss of skeletal muscle function. In T1DM, the deficiency of C-peptide along with insulin is associated with an impairment of skeletal muscle functions such as growth, and repair, and is thought to be an important contributor to increased morbidity and mortality. Recently, cholesterol-lowering drugs (statins) have also been reported to increase the risk of skeletal muscle dysfunction. We hypothesised that C-peptide activates key signaling pathways in myoblasts, thus promoting cell survival and protecting against simvastatin-induced myotoxicity. This was tested by investigating the effects of C-peptide on the L6 rat myoblast cell line under serum-starved conditions. Results: C-peptide at concentrations as low as 0.03 nM exerted stimulatory effects on intracellular signaling pathways—MAP kinase (ERK1/2) and Akt. When apoptosis was induced by simvastatin, 3 nM C-peptide potently suppressed the apoptotic effect through a pertussis toxin-sensitive pathway. Simvastatin strongly impaired Akt signaling and stimulated the reactive oxygen species (ROS) production; suggesting that Akt signaling and oxidative stress are important factors in statin-induced apoptosis in L6 myoblasts. The findings indicate that C-peptide exerts an important protective effect against death signaling in myoblasts. Therefore, in T1DM, the deficiency of C-peptide may contribute to myopathy by rendering myoblast-like progenitor cells (involved in muscle regeneration) more susceptible to the toxic effects of insults such as simvastatin.

## 1. Introduction

In Type 1 diabetes mellitus (T1DM), impaired skeletal muscle health is observed and is termed diabetic myopathy (DMy) [1]. This complication of T1DM is poorly studied but is associated with the progression of other diabetic complications and directly influences the development of co-morbidity, possibly due to the fact that skeletal muscle functions are the largest site for glucose uptake [2]. Therefore, changes to skeletal muscle health may have an impact on whole-body glucose homeostasis [1]. In uncontrolled T1DM, complete insulin deficiency contributes to rapid muscle protein loss [3] due to an imbalance between synthesis and breakdown where protein catabolism exceeds any increase in protein synthesis resulting from the release of amino acids from the degraded protein [4].

There are many possible underlying causes of DMy in T1DM. Hyperglycemia and hypoinsulinemia are obvious candidates, and oxidative stress secondary to hyperglycemia also probably contributes to DMy [1], reinforced by a number of other mechanisms that lead to oxidative stress [5]. High glucose exerts a detrimental effect not only on mature skeletal muscle, but also on satellite cell and myoblast proliferation and differentiation (reviewed in [1]).

The absence of other hormones (including possibly C-peptide) may also be implicated in DMy. Furthermore, DMy may be exacerbated by medications used to treat other conditions in diabetes, such as statins used to treat hyperlipidemia. Lipid-lowering agents, particularly statin inhibitors of 3-hydroxy-3-methylglutaryl coenzyme A (HMG-CoA) reductase have been reported to induce myopathy and lead to muscle weakness [6,7]. Myopathy is a commonly reported side effect of statins [6,7,8,9,10,11] although the magnitude of the effect is controversial. The mechanisms by which statins induce myopathy are not fully understood. Several have been proposed [11] including disruption of mitochondrial activity and induction of apoptosis [6]. These mitochondrial defects may account for the apoptosis, increase mitochondrial reactive oxygen species (ROS), and promote muscle remodeling and degeneration [12,13].

C-peptide is a cleavage product of proinsulin and is secreted in equimolar amounts with insulin. It consists of 31-amino acids and circulates in nanomolar concentrations (typically 0.6 to 3 nM in healthy individuals [14,15]). C-peptide has a longer half-life than insulin and therefore higher concentrations of C-peptide persist in the peripheral circulation. There is now unequivocal evidence that C-peptide exerts biological effects reminiscent of a peptide hormone [15]. In a number of studies, C-peptide has been reported to play protective roles under diabetic conditions. In human endothelial cells, C-peptide significantly reduced glucose-induced apoptosis [16,17]. In a previous study using L6 myoblasts, C-peptide activated ERK1/2, but was reported to show no effect on Akt phosphorylation [18]. In vitro and in vivo studies have investigated the biological effect of C-peptide on muscle glucose utilization and amino acid uptake [19]. However, the possibility of corresponding protective effects of C-peptide on diabetic skeletal muscle (for example, as a potential suppressor of death signaling) has not been investigated. Therefore, the hypothesis proposed here is that C-peptide may represent an important factor in reversing or preventing myopathy (DMy) associated with T1DM and in protecting against potential metabolic insults induced by simvastatin.

## 2. Results 

### 2.1. Effect of Rat C-Peptide on ERK1/2 Activation in L6 Cells

The effect of low concentrations of C-peptide on ERK1/2 activation in L6 cells was studied (Figure 1). Compared to unstimulated controls, there was a detectable increase in ERK1 activation with C-peptide concentrations as low as 0.01 nM, and an apparent dose-dependent increase between 0.01 and 0.1 nM (Figure 1A,B). However, with a higher concentration (1 nM) the activation of P-ERK1 again dropped to control level. A similar biphasic dose dependence of the effect of C-peptide was seen for ERK2 phospho-activation, as shown in Figure 1B. 

### 2.2. C-Peptide Stimulates Akt in L6 Myoblasts

The effects of rat C-peptide on Akt activation in L6 myoblasts were investigated by immunoblot analysis of Akt phosphorylation. Contrary to an earlier report that C-peptide had no effect on Akt phosphorylation at Thr308 in L6 myoblasts [18]. In the present study, rat C-peptide evoked Akt phosphorylation significantly at Ser473 after treating cells for 5 min at doses from 0.01 to 0.3 nM. On stimulation with 0.3 nM rat C-peptide, Akt phosphorylation increased more than 20 fold over the base line (Figure 2A,B). Robust, concentration-dependent activation of Akt by rat C-peptide was observed over the whole dose range from 0.01 nM to 3 nM. To determine whether this C-peptide effect on Akt was sensitive to pertussis toxin (PTX), cells were pre-incubated with 100 ng/ml PTX in serum free medium for 1 h. Pre-treatment of cells with PTX entirely blocked the phosphorylation of Akt induced by 3 nM C-peptide, with Akt activation reduced to control levels (Figure 2A,B). 

### 2.3. C-Peptide Suppresses Simvastatin Induced Death Signaling in Skeletal Muscle Cells 

#### 2.3.1. Simvastatin Effects on Myoblast Viability

To confirm that L6 myoblasts are a suitable model for statin-induced myopathy, the time course and dose dependency of simvastatin effects on L6 myoblast viability were determined from the methylthiazoletetrazolium (MTT)-derived color intensity. Treatment with simvastatin significantly decreased viability of L6 myoblasts in both dose and time dependent manner (Figure 3A). This effect increased in magnitude with an increasing length of simvastatin incubation, reaching statistical significance after 72 h. Similarly increasing concentrations of statin resulted in a progressively reduced cell viability. 

#### 2.3.2. C-Peptide Protects against Simvastatin-Induced Cytotoxicity

To investigate whether rat C-peptide protects L6 cells against simvastatin-induced cell death, L6 myoblasts were incubated in the presence of 3 nM rat C-peptide and simvastatin. A fresh test medium was added every 24 h. The decrease of cell viability induced by simvastatin after 72 h treatment was significantly blunted by the co-application of C-peptide. The protective effect of C-peptide was especially noticeable on incubation with 10 and 30 µM simvastatin (Figure 3B). 

In order to investigate whether the protective effect of rat C-peptide on the simvastatin induced decrease in myoblast viability was a specific G-protein coupled receptor (GPCR)-mediated effect, pertussis toxin (PTX) was used. The results in Figure 3C show that 100 ng/mL PTX alone decreased cell viability slightly. As in Figure 3B, 3 nM rat C-peptide suppressed the toxic effect of simvastatin at 10 and 30 µM, but PTX significantly blunted this protective effect of C-peptide against simvastatin-induced cell toxicity, reducing cell viability to the same level as that seen with simvastatin alone (Figure 3C). Further confirmation of the specificity of the C-peptide effect was obtained by using 3 nM human scrambled C-peptide as a negative control. This scrambled peptide showed no protective effect against the 30 µM simvastatin-induced decrease in cell viability (Figure 3C).

#### 2.3.3. Morphological Visualization

In spite of serum starvation for 72 h, untreated control cultures maintained adherence on the culture plate, with healthy myoblast morphology. Only a small number of pycnotic and damaged cells were observed. (Figure 4(Ai)). However, there was an increasing number of pycnotic and shrunken cells on treating the cultures with 10 µM (Figure 4(Aiii)), indeed some of the cells were found to be losing adherence and floating after 72 h with statin. This effect increased with increasing concentration of simvastatin. After incubation with 3 nM rat C-peptide alone, cells appeared viable and healthy with normal morphology (Figure 4(Aii)). Interestingly, co-incubation of the cells with 3 nM rat C-peptide seemed to blunt the damaging effect of simvastatin at 10 µM at the 72 h time point (Figure 4(Aiv)), allowing the cells to maintain near normal morphology. 

#### 2.3.4. C-Peptide Activates Akt and Suppresses the Inhibitory Effect of Simvastatin on Akt Activation in Myoblasts.

To determine whether the cytotoxic effect of simvastatin on L6 myoblast viability is mediated by the inhibition of Akt, and whether the protective effect of C-peptide is mediated by Akt activation, phospho-activation of Akt (phospho Akt) was investigated by Western blotting. Initially studies were performed at early time points (90 min and 24 h) to determine if simvastatin triggers a potentially pro-apoptotic change in Akt signaling earlier than the 72 h statin-induced damage becomes apparent in cell morphology. Treatment of the cells with 10 or 30 μM simvastatin for 90 min or 24 h showed an inhibitory effect on phospho Akt compared to the baseline untreated cells, and in the presence of 3 nM rat C-peptide this inhibitory effect of simvastatin was suppressed (data not shown). At 72 h, a clear inhibitory effect of simvastatin on phospho Akt was observed with as little as 10 µM simvastatin; and 3 nM rat C-peptide clearly blunted this inhibitory effect of statin on phospho Akt activation (Figure 4B,C). 

#### 2.3.5. C-Peptide Suppresses Simvastatin-Induced Caspase-3 Cleavage

To investigate the possibility that the simvastatin-induced decline in cell viability observed in Figure 4 occurred by induction of apoptosis, caspase-3 cleavage was used as an indicator of apoptosis in L6 myoblasts. 

The clear apparent toxicity effects in the MTT assay (Figure 3) after 72 h incubation with simvastatin were reflected in the caspase-3 experiment in which the 17 kD cleavage fragment increased significantly in cultures incubated with simvastatin for 72 h as shown in (Figure 5A,B) and the protective effect of co-incubation of rat C-peptide with simvastatin was also clearly apparent. Densitometry performed over four experiments showed that this effect of C-peptide was statistically significant when compared to cells treated with 10 µM simvastatin alone (Figure 5A,B).

#### 2.3.6. Simvastatin-Induced Reactive Oxygen Species (ROS) Generation in L6 Myoblasts

It has been reported that simvastatin can induce oxidative stress [5,13] and thus the reactive oxygen species (ROS) may, at least partly, mediate simvastatin-induced myopathy. C-peptide has attracted attention because of its antioxidant properties and ability to protect against high glucose-induced increase in ROS in endothelial cells [17]. Experiments were therefore performed using the ROS probe NBT to evaluate whether the apparent protective effect of rat C-peptide in simvastatin-induced cell death in L6 myoblasts might be mediated through changes in the ROS.

Incubations of L6 of up to a 360 min duration with simvastatin and C-peptide (after 18 h of prior serum starvation) showed clear time-dependent and dose-dependent effects (Figure 6A) on the production of ROS. Compared to untreated cells, the production of ROS was significantly increased in a dose-dependent manner by simvastatin at all time points from 30–360 min. The 100 μM H_2_O_2_ and 30 μM menadione (M) which were used as positive controls also showed a clear increase in ROS production, although this increase apparently faded from 180 min onwards, no longer reaching statistical significance compared with control cells (Figure 6A). 

#### 2.3.7. C-Peptide Reduces Simvastatin-Induced ROS Generation in L6 Myoblasts.

Rat C-peptide at the physiologically relevant concentration of 3 nM decreased simvastatin-induced accumulation of ROS in L6 myoblasts. This effect was detectable after as little as 30 and 60 min of treatment of the cells with simvastatin in combination with 3 nM C-peptide (Figure 6B,C). After 180 and 360 min (Figure 6D,E) of treatment of the cells with C-peptide, this effect was particularly marked; indeed C-peptide attenuated simvastatin-induced ROS generation almost down to that seen in control cultures without simvastatin.

## 3. Discussion

Myopathy or muscle atrophy occurs in a range of pathological states, including diabetes, cancer, uraemia, sepsis, and respiratory insufficiency, and can also be a consequence of aging. Cholesterol-lowering drugs such as statins have also been suggested to be associated with muscle atrophy and myopathy and may worsen age-related muscle wasting (sarcopaenia) [6,7,8,9,10,11,12,13,20]. 

In this work, multiple signaling effects of C-peptide were identified in L6 myoblasts. In healthy fasted humans the plasma concentration of C-peptide is reported to be 0.3–0.6 nM, rising to 1–3 nM after a meal [14]. The present data unambiguously indicate that C-peptide at physiologically relevant concentrations transiently increased ERK1/2 phosphorylation with reproducible stimulatory effects observed at doses as low as 0.03 and 0.1nM. Such concentrations are consistent with the reported affinity of C-peptide for human cell membranes [21]. This could indicate that C-peptide receptors on L6 cell membranes are relatively few and show high-affinity binding, thereby reaching saturation at low C-peptide concentrations. Other investigators have also reported such transient effects of C-peptide on ERK1/2 activation [22]. Likewise, phosphorylation of Akt by C-peptide was observed at 5 min and persisted even at 72 h, providing support for the concept that C-peptide may play a cell survival role through this pathway and protect against death signaling and diabetic complications [15]. This work also provides evidence that C-peptide actions were mediated through a PTX-sensitive G-protein coupled receptor (GPCR). First, the C-peptide stimulatory effect on Akt activation was inhibited by PTX. Second, the protective effect of C-peptide on simvastatin-induced impairment of cell viability was also inhibited by PTX treatment. These effects are consistent with previous reports that C-peptide actions are PTX sensitive [23,24]. The reason for the transience of some of the signals triggered by rat C-peptide (especially on ERK) in L6 myoblasts is unclear. In mammals, the skeletal muscle is thought to be the main contributor, after the kidney, to the clearance of C-peptide from circulation [25]. For that reason the transient effects of C-peptide on ERK signaling observed in L6 myoblasts might arise because of some form of C-peptide inactivation or sequestration, for example peptidase cleavage, or rapid internalisation of the peptide into the cells. However, the observation that, at least in some experiments, C-peptide gave a bell-shaped dose-response curve, with a declining response at higher concentrations, in addition to the transience of the effects, suggests that some form of C-peptide receptor desensitisation occurs in L6 cells. It has previously been suggested that such desensitisation effects may occur in T2DM in which elevated circulating concentrations of C-peptide may be found, thus contributing to an apparent C-peptide resistance in these patients [15]. However, the activation of these signaling pathways by C-peptide in myoblasts raises the possibility that C-peptide could protect against the negative impact of T1DM on skeletal muscle growth and regeneration and decrease the development of diabetic myopathy.

The current data strongly suggest that in L6 myoblasts C-peptide is a potential survival factor as it activated both ERK1/2 and Akt. Akt activation is a well-established cell growth and survival signal [26]. The data demonstrate that simvastatin significantly decreases L6 cell viability. This is consistent with previous work that has demonstrated that simvastatin induces cytotoxicity effects on C2C12 myotubes [20]. In the current study, the observation of a protective effect of rat C-peptide at a physiologically relevant concentration against simvastatin-induced death signaling in L6 myoblasts makes it important to understand which pathway(s) is/are involved. C-peptide’s action seems to be associated with at least two important processes: Stimulation of Akt signaling and suppression of oxidative stress. Previous studies have shown that one of the effects of simvastatin is to inhibit Akt/FoxO signaling in cancer cell lines and in C2C12 skeletal muscle cells, thus promoting apoptosis [26]. Agents that strongly stimulate Akt signaling would therefore be expected to reverse this pro-apoptotic effect of simvastatin. For example, the insulin-like growth factor-I (IGF-I) has been reported to protect against simvastatin-induced C2C12 damage at 24h and the protective effect was shown to be mediated by Akt [20]. Similarly, in the present study C-peptide activates Akt within 5 min in L6 myoblasts, suggesting that a similar protective effect against simvastatin, through an Akt-dependent pathway, may also occur with C-peptide. The present results show that C-peptide does indeed suppress the inhibitory effect of simvastatin on Akt phosphorylation, and this protective effect on Akt stimulation persisted even in incubations with simvastatin and C-peptide lasting as long as 72 h. The observation that simvastatin strongly inhibits activation of Akt, and that C-peptide reverses this signaling defect, strongly suggests that changes in Akt signaling play a major role in the protective effect of C-peptide in the face of simvastatin-induced apoptosis. To prove more rigorously the functional importance of this C-peptide effect on Akt, it would be of interest in the future to compare with the present results the effect of C-peptide on apoptosis in L6 cells in which apoptosis has been induced by some irreversible inhibition of Akt which cannot readily be over-ridden by C-peptide, for example by Akt gene-silencing with siRNA or by direct pharmacological inhibition of Akt [27]. In addition, to the evidence of apoptosis from the changes in morphology, simvastatin was shown to induce caspase-3 activation. This was indicated by the accumulation of the 17 kD cleavage product. C-peptide inhibited this activation of caspase-3, analogous to previous work which indicated that C-peptide inhibits caspase-3 activation induced by high glucose in endothelial cells [16]. 

In addition, to simvastatin-induced inhibitory effects on Akt and their reversal by C-peptide, it was also clear from the present data that the ROS generation by simvastatin, and its prevention by C-peptide, were likely contributors to the effects of simvastatin and C-peptide on cell survival. The molecular basis of these effects on ROS production, and the precise relationship between the Akt and ROS effects of simvastatin in L6 myoblasts, and their suppression by C-peptide, are at present unknown. By analogy with the effects of C-peptide that have been reported in vascular endothelial cells [17], C-peptide signaling through its hypothetical GPCR may directly inhibit the production of ROS by inhibiting the normal Rac-1 (Rho GTPase)-dependent activation of NAD(P)H oxidase [17,28]. Whether Akt also has any influence on this process in L6-myoblasts remains to be determined. In T1DM, the deficiency of insulin and/or C-peptide as potential survival factors may play a role in the pathogenesis of myopathy by rendering skeletal muscle more susceptible to the toxic effects of insults such as simvastatin. The observations in this work strongly support a role for C-peptide in promoting survival of skeletal muscle myoblasts in the context of statin therapy in T1DM.

## 4. Material and Methods 

### 4.1. Chemical and Reagents

Rat C-peptide and a 31-amino acid scrambled C-peptide were provided by John Wahren (Karolinska Institute, Stockholm, Sweden) and were dissolved in 0.9% *w*/*v* NaCl. Simvastatin was obtained from Sigma-Aldrich (Gillingham, Dorset, UK) and dissolved in dimethyl sulfoxide (DMSO). Pertussis toxin (PTX) was obtained from (Merck, Nottingham, UK).

### 4.2. Cell Culture

L6 myoblasts (sub-clone G8C5) [29] were obtained from the European Collection of Animal Cell Cultures (Ref. 92121114) and were used between passages seven and 19. Routine propagation of L6 cells was performed in Growth Medium comprising Dulbecco’s Modified Eagle’s Medium (DMEM) (Invitrogen 11880, low glucose) supplemented with batch-tested 10% *v*/*v* heat-inactivated foetal bovine serum (FBS; Invitrogen (Life Technologies), Paisley, Scotland, UK), 100 U/mL Penicillin G, 100ug/ml Streptomycin (Invitrogen 15140), 2mM l-glutamine (Invitrogen 25030) and 10mg/L Phenol Red (Lutterworth, Leicestershire, UK). Cells were serum starved overnight before incubation with C-peptide. The other components of the test media were varied depending on the experimental conditions required and are described in detail in the figure legends in the results section.

### 4.3. Immunoblotting

Cell lysates were subjected to SDS-PAGE (30 µg protein per lane) then transferred onto nitrocellulose membranes (Amersham/GE Healthcare Hybond, Ref RPN203D) followed by probing with primary antibodies against Thr202/Tyr204 p-ERK1/2, total ERK2, Ser473 p-Akt, total Akt (Cell Signalling Technology, London, UK); Caspase 3 (Santa Cruz-Insight Biotechnology, Wembley, UK) and Actin (C-terminal anti-Actin antibody, Clone AC-40 (Sigma-Aldrich, A4700, Gillingham, Dorset, UK)). Polyclonal goat anti-rabbit or rabbit anti-mouse IgG/HRP (Dako Cytomation, Glostrup, Denmark) were used as secondary antibodies as appropriate, and HRP-labelled proteins were detected by chemiluminescence (ECL reagent) from (Bio-Rad, Watford, UK). Band intensities were quantified by the Image Studio Software v 4.0.21 (LI-COR Biosciences, Lincoln, NE, USA), and data are presented as the ratio of the intensity for the protein of interest/housekeeping protein.

### 4.4. Cell Viability and Growth 

Cell viability was measured by the methylthiazoletetrazolium (MTT) assay as described previously [30].

### 4.5. Wright Stain

Myoblasts that had been treated with experimental test media were stained by using the Wright stain [31] (Sigma-Aldrich, Gillingham, Dorset, UK). Wright staining was performed to detect characteristic apoptotic features including chromatin condensation, cell shrinkage, membrane blebbing, and membrane-bound apoptotic bodies.

### 4.6. Determination of ROS Generation

The generation of ROS by myoblast cultures were assessed colorimetrically using the water-soluble dye Nitro Blue Tetrazolium (NBT) (Sigma Ref 74032) as previously described [32]. The principle of the assay is that NBT is reduced by superoxide radicals to a blue, water-insoluble formazan product [32].

### 4.7. Statistical Analysis

Data are presented as the mean ± SEM and were analysed using the GraphPad Prism 6.0 (US). Data normality was checked with the Kolmogorov-Smirnov test. Two group data comparisons were analysed by the t test (for normally distributed data) or by Wilcoxon matched-pairs signed rank test (for nonparametric data). One-way ANOVA (combined with Tukey’s post hoc test (for normally distributed data) or Dunn’s (nonparametric) post hoc test was applied for multiple comparison tests as appropriate. For experiments in which data were normalized and expressed as a ratio (sample/control), data were log transformed before analysis. The “n” values denote the number of independent experiments. *p* < 0.05 was considered statistically significant.

## 5. Conclusions

In view of the important functional effects of C-peptide on myoblasts reported in this study, it seems feasible that C-peptide might protect against the development of myopathy arising from damaging effects on myoblast-like satellite cells in T1DM patients who receive long-term treatment with statins. Thus, C-peptide may suppress the negative impact of T1DM and statins on muscle regenerative potential. It will be of future interest to see whether statins and C-peptide exert effects like those reported here on cultured human myoblasts grown from normal and T1DM skeletal muscle biopsies; and in the longer term to determine whether C-peptide administered in T1DM can exert beneficial effects on skeletal muscle satellite and progenitor cell biology in vivo. 

Finally, it should be noted however, that controversy has surrounded the most appropriate doses of simvastatin to use for in vitro cell culture studies. The doses of simvastatin for the present study were chosen to be in line with earlier published work that demonstrated that simvastatin induces cytotoxicity effects on C2C12 myotubes and decreases the ubiquinone concentration (CoQ10 levels) [20]. However, while some authors have believed that the plasma concentration of simvastatin in the human blood may reach approximately 4 to 5 μM at a dose of 40 mg simvastatin/day, it has also been reported that the plasma concentrations at this dose may only reach maximal values close to 100 nM ([33]). A further complication is that simvastatin itself is only a pro-drug and ideally the concentration of the pharmacologically active hydroxy acid metabolite needs to be known [34] The interaction between C-peptide and simvastatin therefore merits further investigation at lower doses, particularly in view of the report that repeated exposure of cells to low doses of simvastatin can exert a significant cytostatic effect, thus potentially impairing tissue regeneration [35]. 

## Figures and Tables

**Figure 1 ijms-20-01654-f001:**
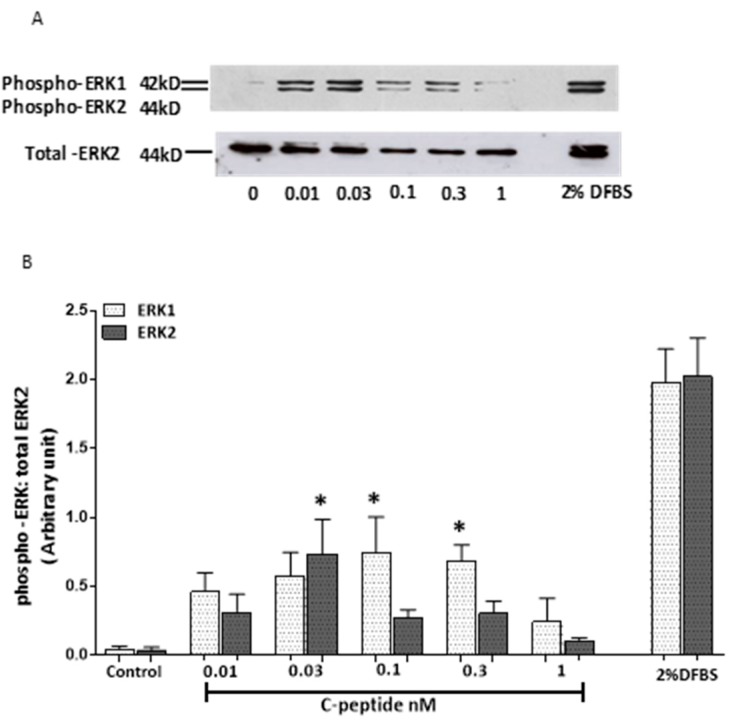
Rat C-peptide activates ERK1/2 in rat skeletal muscle cell line L6. L6 myoblasts cells were stimulated with rat C-peptide for 5 min in Dulbecco’s Modified Eagle’s Medium (DMEM). (**A**) Phosphorylation of ERK1/2 was determined by the Western blot using specific anti-phospho ERK1/2 antibody. 2% dialysed foetal bovine serum (DFBS) was used as a positive control. As a loading control, membranes were reprobed with an antibody against ERK2. Quantification by densitometry of data from three such experiments is presented in histograms; (**B**) as mean ± SEM. * *p* < 0.05 versus the unstimulated control.

**Figure 2 ijms-20-01654-f002:**
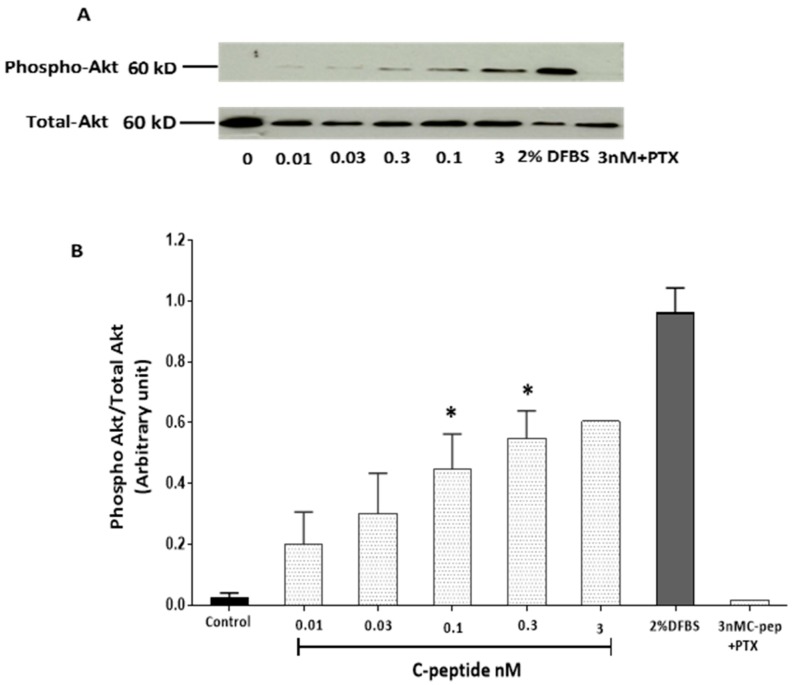
Dose response study of Akt activation by rat C-peptide in L6 cells. Cells were stimulated with rat C-peptide for 5 min in DMEM. Pertussis toxin (PTX) denotes the effect of co-incubation with 100 ng/mL Pertussis toxin. Phosphorylation of Akt was determined by the Western blot (**A**) using specific anti-phospho-Akt antibody. DFBS (2%) was used as a positive control. As a loading control, membranes were reprobed with antibody against total Akt; (**B**) quantification by densitometry of data from three such experiments: mean ± SEM * *p* < 0.05 versus the unstimulated control.

**Figure 3 ijms-20-01654-f003:**
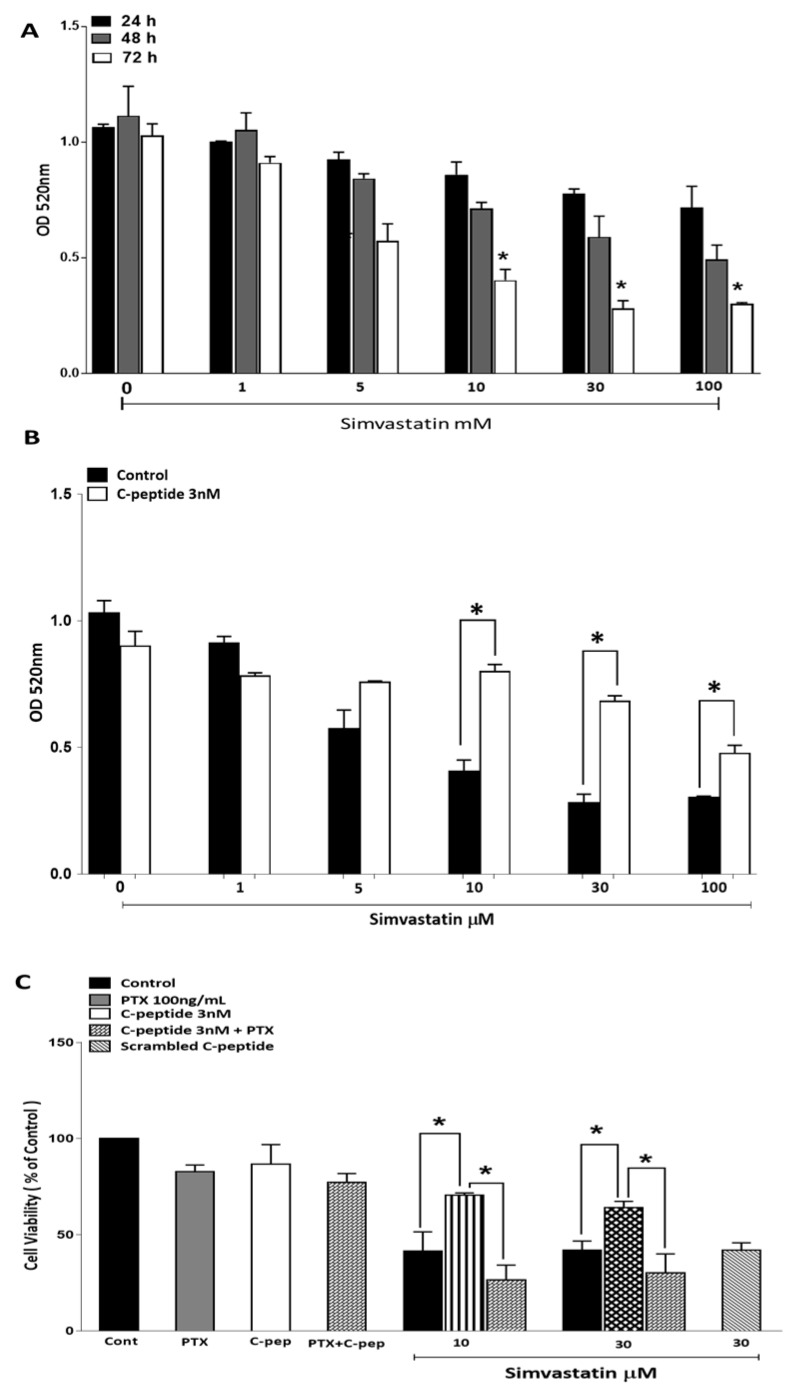
Simvastatin decreases L6 myoblast viability in a time- and dose-dependent manner (**A**). Cytotoxicity was assessed by methylthiazoletetrazolium (MTT) assay in cells treated with simvastatin at different concentrations and at different time points indicated above. Data are mean ± SEM of (*n* = 4) independent experiments performed in triplicate, * *p* < 0.05 (compared to untreated cells at the same time point); (**B**) C-peptide protects against simvastatin-induced cytotoxicity assessed using MTT assay. Cells were incubated with varying concentrations of simvastatin (with or without 3 nM C-peptide) for 72 h. Data are presented as mean ± SEM of four independent experiments with each condition performed in triplicate, * *p* < 0.05; (**C**) pertussis toxin, PTX (100 ng/mL) inhibits the protective effect of C-peptide against simvastatin-induced cell death. Cytotoxicity was assessed using the MTT assay. Cells were treated for 72 h as indicated in the key on the figure. 3 nM scrambled human C-peptide was used as a negative control. Data are presented as % of the untreated control value (Cont) in each experiment. Values are mean ± SEM of three independent experiments performed in duplicate, * *p* < 0.05.

**Figure 4 ijms-20-01654-f004:**
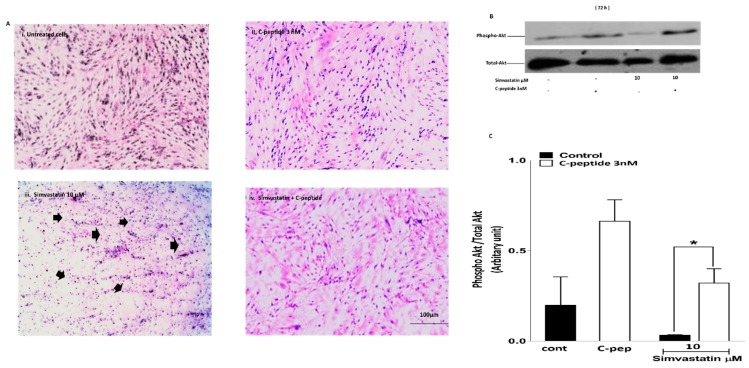
(**A**) C-peptide protects against simvastatin-induced cell toxicity. L6 myoblasts were serum starved in DMEM (**i**); treated with 3 nM C-peptide alone (**ii**); treated with simvastatin 10 μM (**iii**); or treated with simvastatin in the presence of 3 nM rat C-peptide (**iv**) for 72 h. Morphological visualization was assessed by using the Wright stain and light microscopy. Black arrows indicate shrunken cells (consistent with apoptosis). Magnification ×200; (**B**) C-peptide blunts the inhibitory effect of simvastatin on phospho-Akt activation in L6 myoblasts. Cells were treated with 10 µM simvastatin and co-incubated with 3 nM rat C-peptide for 72 h. Phosphorylation of Akt was determined by the Western blot using specific anti-phospho Akt antibody. As a loading control, membranes were reprobed with antibody against total Akt (shown in the lower blot); (**C**) densitometric analysis of four experiments. Data are presented as mean ± SEM * *p* < 0.05.

**Figure 5 ijms-20-01654-f005:**
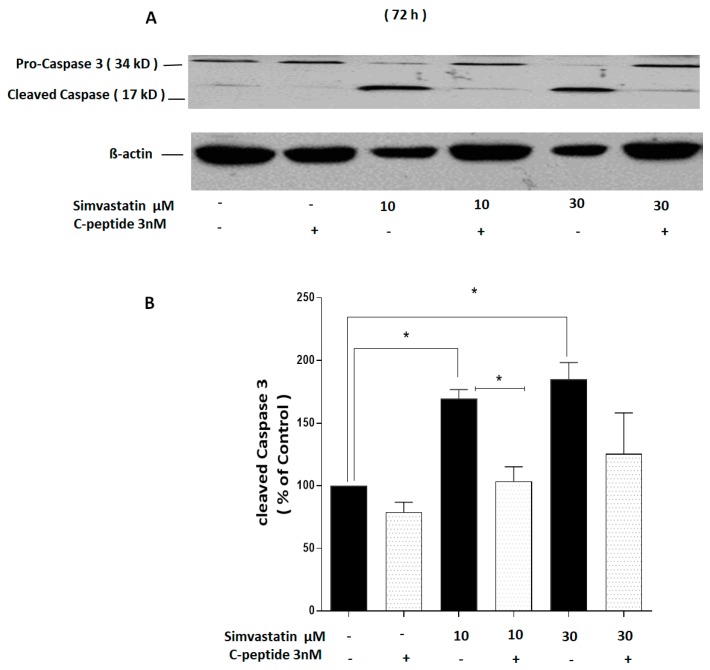
C-peptide blunts the effect of simvastatin–induced caspase-3 cleavage in L6 myoblasts. Cells were treated with 10 and 30 µM simvastatin and co-treated with 3 nM rat C-peptide for 72 h. (**A**) Cleavage of caspase-3 was determined by the Western blot using specific anti-caspase-3 antibody. As a loading control, membranes were reprobed with antibody against β actin; (**B**) densitometry analysis of four experiments was performed at 72 h and data are presented as mean ± SEM.* *p* < 0.05.

**Figure 6 ijms-20-01654-f006:**
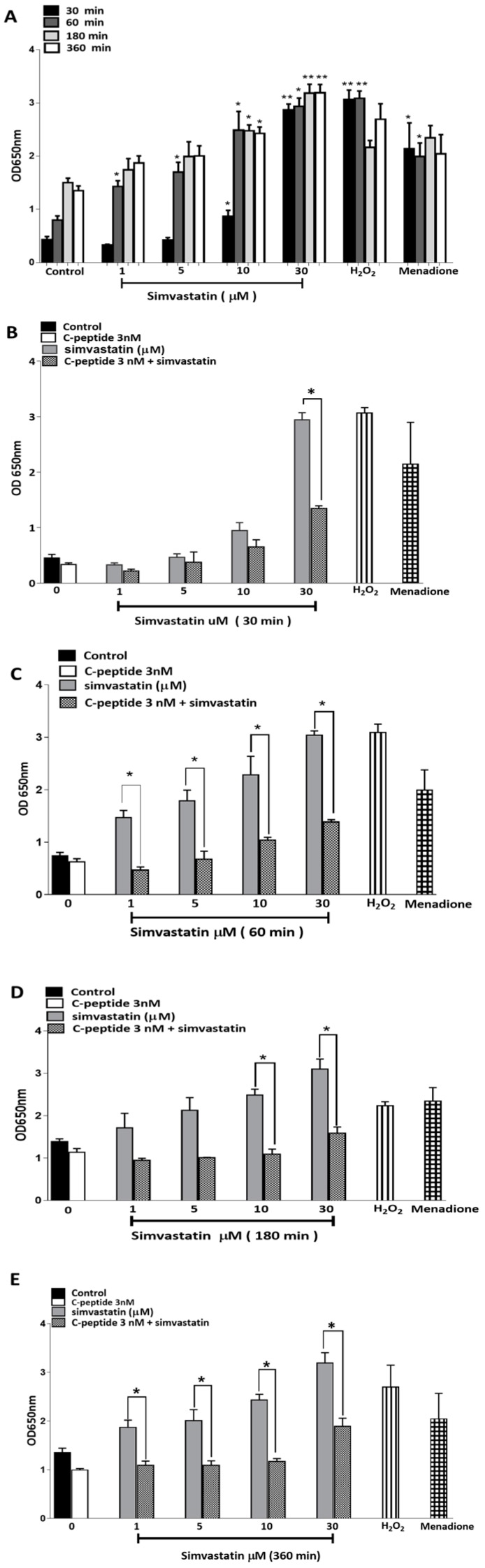
Simvastatin-induced ROS generation in L6 myoblasts, assessed by colorimetric detection using Nitro Blue Tetrazolium (NBT). (**A**) Cells were serum starved overnight in DMEM and then treated with a range of simvastatin concentrations for the times shown. H_2_O_2_ (100 μM) and (M) menadione (30 μM) were used as positive controls. Data are pooled from three independent experiments and presented as mean ± SEM. * *p* < 0.05; ** *p* < 0.01 compared to control at the same time point. (**B**–**E**) C-peptide blunting of simvastatin-induced ROS generation in L6 myoblasts. Cells were serum starved overnight in DMEM and treated with simvastatin alone or in-combination with 3 nM C-peptide for 30–360 min. H_2_O_2_ (100 μM) and menadione (M) (30 μM) were used as positive controls. Data are pooled from three independent experiments and presented as mean ± SEM. * *p* < 0.05.

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
