# Peer review of "Proinsulin C-Peptide Enhances Cell Survival and Protects against Simvastatin-Induced Myotoxicity in L6 Rat Myoblasts"

_ijms, 2019, doi:10.3390/ijms20071654_

Round 1

Reviewer 1 Report

In this work, multiple signalling effects of C-peptide were identified in L6 rat myoblast cell line under serum-starved conditions. C-peptide at concentrations as low as 0.03nM exerted stimulatory effects on intracellular signalling pathways - MAP kinase (ERK1/2) and Akt. When apoptosis was induced by simvastatin (10-100μM), 3nM C-peptide potently suppressed the apoptotic effect through a pertussis toxin-sensitive pathway. Authors performed the immunoblotting, Wright stain and colorimetric detection.

Authors determined that simvastatin strongly impaired Akt signalling and stimulated reactive oxygen species (ROS) production; suggesting that Akt signalling and oxidative stress are important factors in statin-induced apoptosis in L6 myoblasts. They concluded C-peptide exerts an important protective effect against death signalling in myoblasts.

Subject definitions and methods are optimal. Figures and references are adequate.

I have found this paper relevant to the field of this journal. I have only one minor comment.      

Minor point:

1) In the Figure 6, there are missing parts B-E. They are mentioned in the text and in the legend, but they are not shown as graphs.

2) The Paragraph 3.3.2, the row 176, the first you should give the full name of effect and then its abbreviation in parenthesis - G-protein coupled receptor (GPCR)-mediated effect, …

I recommend this paper for acceptation after minor revision in the journal.

Author Response

Response to Reviewer 1 Comments

Reviewer 1. In this work, multiple signalling effects of C-peptide were identified in L6 rat myoblast cell line under serum-starved conditions. C-peptide at concentrations as low as 0.03nM exerted stimulatory effects on intracellular signalling pathways - MAP kinase (ERK1/2) and Akt. When apoptosis was induced by simvastatin (10-100μM), 3nM C-peptide potently suppressed the apoptotic effect through a pertussis toxin-sensitive pathway. Authors performed the immunoblotting, Wright stain and colorimetric detection. Authors determined that simvastatin strongly impaired Akt signalling and stimulated reactive oxygen species (ROS) production; suggesting that Akt signalling and oxidative stress are important factors in statin-induced apoptosis in L6 myoblasts. They concluded C-peptide exerts an important protective effect against death signalling in myoblasts.  Subject definitions and methods are optimal. Figures and references are adequate.I have found this paper relevant to the field of this journal. I have only one minor comment.      

We thank the reviewer for these helpful comments. Our responses to the specific points made are as follows:

1)    In Figure 6, there are missing parts B-E. They are mentioned in the text and in the legend, but they are not shown as graphs.

We thank the reviewer for pointing out this error which has now been corrected. The missing parts of Figure 6 have now been inserted.

     2) The Paragraph 3.3.2, the row 176, the first you should give the full name of effect and then its abbreviation in parenthesis - G-protein coupled receptor (GPCR)-mediated effect, …

We apologise for the omission of the definition of GPCR. The definition has now been inserted in the text.

Reviewer 2 Report

In the submitted study, the authors identified a novel protective role of C-peptide against simvastatin-induced myotoxicity in L6 rat myoblasts. By suppressing the inhibitory effect of simvastatin on Akt activation and reducing simvastatin-induced ROS production, C-peptide effectively enhanced cells survival and blunted the apoptotic effect on L6 myoblasts in the context of simvastatin treatment. The findings may provide a novel mechanistic insight into the protective effect of C-peptide on T1DM associated myopathy. However, I have a few suggestions for improvement.

1)     The font size for words and numbers in Figure 3, Figure 4 and Figure5 are too small and unclear to read. Please change for clear and larger font size.

2)     Please change Figure 4A for pictures with color. Label each picture with treatment and point out the characteristic apoptotic features by arrow.

3)     In Figure 5, the treatment description in 5A and 5B are not consistent. Please correct the description for lane 2 and rearrangement the description, for example put “simvastatin” above “C-peptide” in both 5A and 5B.

4)     In Figure 4B, the expression of total Akt varies a lot, please provide a loading control.

5)     In Figure 5 legend line 233-235, the author mentioned that “H2O2 (100 μM) was used as positive control …and 1% DFBS was used to assess….serum starvation”. There were no data showed in the figures. Please provide figures including these two treatments.

6)     In Figure 1, 2 and 4, 2% DFBS was used as positive control. However, in Figure 3 and 5, 1% DFBS was used as control. Please explain the reason for using different concentrations of DFBS as control.

7)     Figure 6B-E are missing. Please correct.

8)     In the section of Material and Methods, please keep the writing format of each subtitle consistent. For example, capitalize the first letter of every word or only the first word.

9)     In line 97-101, Wright stain was described as a method to assess the characteristic apoptotic features of myoblasts. However, Wright stain is usually used for the differentiation of blood cell types. Please provide references to support the feasibility of this approach in assessing apoptotic features in cells.

10)  In line 102-104, please add description to the NBT assay.

11)  In line 229, the authors described that “the effect of C-peptide was statistically significant when compared to cells treated with 30 μM simvastatin alone”. However, in Figure 5A-B, the significant difference was between cells treated with 10 μM simvastatin.

12)  In Figure 6A, the ROS production induced by H2O2 faded from 180 min. Please explain the possible reason.

13)  In line 280-282, the authors attributed the transient effects of C-peptide on ERK1/2 activation to C-peptide receptors saturation. So how to understand the effects of C-peptide on Akt activation persisted even at 72 h?

14)  Please keep writing format consistent. Fox example, in line 134, 137, 153 190, 194..., add space between numbers and units; in Figure 1B, capitalize the first letter of every word; in line 146, p should be uncapitalized and in italic; in line 159 and Figure 3B, please correct *p<0.01 to *p<0.05 or **p<0.01.

Author Response

Journal

IJMS (ISSN 1422-0067)

Manuscript ID

Ijms-458417

Type Article

Title: Proinsulin C-peptide enhances cell survival and protects against simvastatin-induced myotoxicity in L6 rat myoblasts

Authors

Sumia Essid *, Alan Bevington, Nigel Brunskill

Response to Reviewer 2 Comments

Reviewer 2.

In the submitted study, the authors identified a novel protective role of C-peptide against simvastatin-induced myotoxicity in L6 rat myoblasts. By suppressing the inhibitory effect of simvastatin on Akt activation and reducing simvastatin-induced ROS production, C-peptide effectively enhanced cells survival and blunted the apoptotic effect on L6 myoblasts in the context of simvastatin treatment. The findings may provide a novel mechanistic insight into the protective effect of C-peptide on T1DM associated myopathy. However, I have a few suggestions for improvement.

We thank the reviewer for these helpful comments. Our responses to the specific points made are as follows:

1)     The font size for words and numbers in Figure 3, Figure 4 and Figure5 are too small and unclear to read. Please change for clear and larger font size.

           The font size has now been increased in Figure 3, Figure 4 and Figure 5.

1)    Please change Figure 4A for pictures with color. Label each picture with treatment and point out the characteristic apoptotic features by arrow.

Pictures with colour have been inserted and labelled as recommended by the reviewer.

2)    In Figure 5, the treatment description in 5A and 5B are not consistent. Please correct the description for lane 2 and rearrangement the description, for example put “simvastatin” above “C-peptide” in both 5A and 5B.

We thank the reviewer for pointing out the error and apologise for any confusion caused.

These requested changes have now been made.

3)    In Figure 4B, the expression of total Akt varies a lot, please provide a loading control.

The reviewer has raised a valid point by noting that 30mM simvastatin seems to be producing an anomalous suppression of total Akt, thus tending to increase the observed phospho-Akt/Total Akt ratio. In spite of this, a clear and reproducible overall decline in the phospho-Akt/Total Akt ratio was still observed when compared with the control conditions i.e. the effect that we were reporting was an under-estimate. However, as the reason for this anomalous suppression of total Akt by 30mM simvastatin is at present unknown, and as 10mM simvastatin adequately shows the effect that we are reporting, the data on 30mM simvastatin have now been removed from the figure.

5)     In Figure 5 legend line 233-235, the author mentioned that “H2O2 (100 μM) was used as positive control …and 1% DFBS was used to assess….serum starvation”. There were no data showed in the figures. Please provide figures including these two treatments.

We thank the reviewer for pointing out this typographical error. The original statement was incorrect and has now been removed from the Figure 5 legend.

6)     In Figure 1, 2 and 4, 2% DFBS was used as positive control. However, in Figure 3 and 5, 1% DFBS was used as control. Please explain the reason for using different concentrations of DFBS as control.

Again we thank the reviewer for pointing out these typographical errors. There are no serum controls presented in Figures 3, 4 and 5, and the legends have now been corrected to reflect this.

7)     Figure 6B-E are missing. Please correct.

We thank the reviewer for pointing out this error which has now been corrected. The missing parts of Figure 6 have been inserted.

8)     In the section of Material and Methods, please keep the writing format of each subtitle consistent. For example, capitalize the first letter of every word or only the first word.

More consistent use of capitalization has now been made.

9)     In line 97-101, Wright stain was described as a method to assess the characteristic apoptotic features of myoblasts. However, Wright stain is usually used for the differentiation of blood cell types. Please provide references to support the feasibility of this approach in assessing apoptotic features in cells.

Wright stain has previously been validated for morphological assessment of apoptosis. (Please see for example Gentil, B, Grimot, F & Riva, C. Commitment to apoptosis by ceramides depends on mitochondrial respiratory function, cytochrome c release and caspase-3 activation in Hep-G2 cells Molecular and Cellular Biochemistry 254: 203–210, 2003).

10)  In line 102-104, please add description to the NBT assay.

An explanatory sentence has now been added to Section 5.6.

11)  In line 229, the authors described that “the effect of C-peptide was statistically significant when compared to cells treated with 30 μM simvastatin alone”. However, in Figure 5A-B, the significant difference was between cells treated with 10 μM simvastatin.

We are grateful for this correction.  The text has now been changed to read “10μM simvastatin”.  

12)  In Figure 6A, the ROS production induced by H2O2 faded from 180 min. Please explain the possible reason.

We have not investigated this directly, but a probable explanation is the rapid decomposition of H2O2 by catalase which is known to be strongly expressed in L6 cells (Hidalgo, M et al Cell Physiol Biochem 2014;33:67-77 DOI: 10.1159/000356650).

13)  In line 280-282, the authors attributed the transient effects of C-peptide on ERK1/2 activation to C-peptide receptors saturation. So how to understand the effects of C-peptide on Akt activation persisted even at 72 h?

The reviewer has raised an important point. Why ERK and Akt activation profiles and kinetics differ is unclear - but this has previously been well documented in mammalian cells. Unfortunately a full understanding of the kinetics of C-peptide responses will only be feasible when the receptor has been cloned and fully characterised. It is not known whether there is any heterogeneity of C-peptide receptor sub-types expressed in L6 cells and whether distinct receptor types are signalling to ERK1/2 and Akt.

14)  Please keep writing format consistent. Fox example, in line 134, 137, 153 190, 194..., add space between numbers and units; in Figure 1B, capitalize the first letter of every word; in line 146, p should be uncapitalized and in italic; in line 159 and Figure 3B, please correct *p<0.01 to *p<0.05 or **p<0.01.Please correct this as requested.

These changes have been made as requested.

Reviewer 3 Report

In this study Essid et al have aimed to study the possible role of C-peptide as promoter of skeletal muscle cell survival. Using in vitro techniques on serum-starved L6 rat myoblasts, they found that physiological C-peptide concentrations protect against simvastatin-induce apoptosis. The authors suggest that the mechanism mediating this involve GPCR, Akt and ERK signaling. The manuscript is well written and easy to follow. The study is well conducted, however the lack of a considerable part of the data and the poor resolution of western blot analysis didn’t help to clarify the mechanism. Thus, some questions need to be addressed:

Major:

1- Figure 1. The western blot signal at 0 and 1nM concentration seems to be fading more than being decreased. Besides this, the resolution of the gel is not suitable for analyzing ERK1 and ERK2 independently. Please consider to repeat the WB in a more concentrate SDS-PAGE gel, so the bands can be separated. Why the authors only analyze ERK2 non-phosphorylated protein levels and not ERK1/2 total levels? Please perform a new analysis to extract appropriated conclusions.

2- How the authors explain the different Akt regulation by C-peptide? Did the authors attempt to analyze Thr308-pAkt in this model or any upstream and downstream effector?

3- Page 7 line 180. “…or possibly an even lower level”. This is not correct, the data show no differences between simvastatin alone and simvastatin + c-peptide + PTX groups.

4- Page 8 line 217 (figure 4B and 4C). “At 72h a clear inhibitory effect of simvastatin on phospho Akt”. This is not correct. Seeing the image that the authors showed the ratio phospho Akt/Akt level should be clearly lower in the controls than at 10 and 30μM. No significance is shown either between the controls and simvastatin treated cells. Please show the original blots from 3 experiments and choose appropriate representative images.

5- Figure 6 B-E is missing.

Minor:

-Materials and methods: Please specify the antibody for β-actin protein.

-Please include the legend for each graph bar in the figures (e.g. Control bar group in Figure 3B)

-Figure 3C. The color of the bar of group simvastatin 10μM and 30μM is the same as the untreated Controls and it’s kind of confusing. Please consider to change the color of the bar graph.

-Figure 3, 4 and 5. No DFBS group is shown in this figure but is stated in the figure legend.

-Figure 5. Please check the legend for the treatment in Fig 5A. The group simvastatin-/C peptide + is labelled wrongly.

-Page 8 line 229: “…with 30 μM simvastatin…” Did the authors mean 10μM?

-Page10 line 282. Please check reference 26. In that paper there isn’t any data regarding ERK1/2.

-Page 11 line 301. “…against the negative impact of TIDM on skeletal muscle…” Did the authors mean T1DM?

Author Response

Journal

IJMS (ISSN 1422-0067)

Manuscript ID

Ijms-458417

Type Article

Title: Proinsulin C-peptide enhances cell survival and protects against simvastatin-induced myotoxicity in L6 rat myoblasts

Authors

Sumia Essid *, Alan Bevington, Nigel Brunskill

Response to Reviewer 3 Comments

Reviewer 3.

Comments and Suggestions for Authors

In this study Essid et al have aimed to study the possible role of C-peptide as the promoter of skeletal muscle cell survival. Using in vitro techniques on serum-starved L6 rat myoblasts, they found that physiological C-peptide concentrations protect against simvastatin-induce apoptosis. The authors suggest that the mechanism mediating this involve GPCR, Akt and ERK signaling. The manuscript is well written and easy to follow. The study is well conducted, however the lack of a considerable part of the data and the poor resolution of western blot analysis didn’t help to clarify the mechanism. Thus, some questions need to be addressed:

We thank the reviewer for these helpful comments. Our responses to the specific points made are as follows:

Major:

1- Figure 1. The western blot signal at 0 and 1nM concentration seems to be fading more than being decreased. Besides this, the resolution of the gel is not suitable for analyzing ERK1 and ERK2 independently. Please consider to repeat the WB in a more concentrate SDS-PAGE gel, so the bands can be separated. Why the authors only analyze ERK2 non-phosphorylated protein levels and not ERK1/2 total levels? Please perform a new analysis to extract appropriated conclusions.

The reviewer is correct to be cautious about the apparent “bell-shaped” dose-response curve for C-peptide on phospho-ERK, and we thank the reviewer for these helpful technical suggestions. However, a significant response at low C-peptide doses, accompanied by lack of response at high doses (as apparently observed here) is typical behaviour in a number of cell types and this behaviour is consistent with G-protein coupled receptors of the type that are thought to mediate C-peptide signalling   (Hills, CE & Brunskill NJ, The Review of Diabetic Studies Vol 6 No 3 Special Issue 2009 DOI 10.1900/RDS.2009.6.138)

2- How the authors explain the different Akt regulation by C-peptide? Did the authors attempt to analyze Thr308-pAkt in this model or any upstream and downstream effector?

The reviewer has raised an important point. Why ERK and Akt activation profiles and kinetics differ is unclear - but this has previously been well documented in mammalian cells. Unfortunately a full understanding of the kinetics of C-peptide responses will only be feasible when the receptor has been cloned and fully characterised. It is not known whether there is any heterogeneity of C-peptide receptor sub-types expressed in L6 cells and whether distinct receptor types are signalling to ERK1/2 and Akt.

The suggestion of comparison with Thr308-pAkt is a good one. Ser473-pAkt was studied here because a preliminary experiment with antibody recognising Thr308-pAkt yielded no response after 24 hours of exposure to 3 nM C-peptide in these cells.

3- Page 7 line 180. “…or possibly an even lower level”. This is not correct, the data show no differences between simvastatin alone and simvastatin + c-peptide + PTX groups.

This is a fair point. This phrase has therefore been omitted.

4- Page 8 line 217 (figure 4B and 4C). “At 72h a clear inhibitory effect of simvastatin on phospho Akt”. This is not correct. Seeing the image that the authors showed the ratio phospho Akt/Akt level should be clearly lower in the controls than at 10 and 30μM. No significance is shown either between the controls and simvastatin treated cells. Please show the original blots from 3 experiments and choose appropriate representative images.

The reviewer has raised a valid point.  Simvastatin at 30mM seems to be producing an anomalous suppression of total Akt, thus tending to increase the observed phospho-Akt/Total Akt ratio. In spite of this, a clear and reproducible overall decline in the phospho-Akt/Total Akt ratio was still observed when compared with the control conditions i.e. the effect that we were reporting was an under-estimate. However, as the reason for this anomalous suppression of total Akt by 30mM simvastatin is at present unknown, and as 10mM simvastatin adequately shows the effect that we are reporting, the data on 30mM simvastatin have now been removed from the figure.

5- Figure 6 B-E is missing

We thank the reviewer for pointing out this error which has now been corrected. The missing parts of Figure 6 have been inserted.

Minor:

-Materials and methods: Please specify the antibody for β-actin protein.

The antibody has now been specified in the Material and Methods section.

-Please include the legend for each graph bar in the figures (e.g. Control bar group in Figure 3B)

This Control bar has now been inserted on the figure in Figure 3.

-Figure 3C. The color of the bar of group simvastatin 10μM and 30μM is the same as the untreated Controls and it’s kind of confusing. Please consider to change the color of the bar graph.

Two new colours have been inserted in Figure 3C for the “C-peptide 3 nM + 10 mM Simvastatin” bar and for the “C-peptide 3 nM + 30 mM Simvastatin” bar.

-Figure 3, 4 and 5. No DFBS group is shown in this figure but is stated in the figure legend.

We thank the reviewer for pointing out these typographical errors. There are no DFBS controls presented in Figures 3, 4 and 5, and the legends have now been corrected to reflect this.

-Figure 5. Please check the legend for the treatment in Fig 5A. The group simvastatin-/C peptide + is labelled wrongly.

Again we thank the reviewer for pointing out this typographical error which has now been corrected.

-Page 8 line 229: “…with 30 μM simvastatin…” Did the authors mean 10μM?

Thank you – yes it should be 10μM and has now been corrected..

-Page10 line 282. Please check reference 26. In that paper there isn’t any data regarding ERK1/2.

Thank you for pointing out the error.  An appropriate reference has now been inserted as follows:.

Takanori,K, Kazuhiro, K, JUNG,BD, MAKONDO, K, OKAMOTO, S,  CAN4AS, X, SAKANE, N, YOSHIDA, T, and SAITO, M. (2001 ). Proinsulin C-peptide rapidly stimulates mitogen-activated protein kinases in Swiss 3T3 fibroblasts: requirement of protein kinase C, phosphoinositide 3-kinase and pertussis toxin-sensitive G-protein. Journal of Biochem. J.123–129

-Page 11 line 301. “…against the negative impact of TIDM on skeletal muscle…” Did the authors mean T1DM?

Thank you for pointing out this mistake. Yes it should say T1DM not TIDM, and this change has now been made throughout the manuscript.

Round 2

Reviewer 3 Report

The authors have addressed all my comments